# Comparison of Endoscopic Submucosal Dissection and Radical Surgery for Early Gastric Cancer in Remnant Stomach

**DOI:** 10.3390/jcm11185403

**Published:** 2022-09-14

**Authors:** Yi Liu, Zhihao Chen, Hong Zhou, Yingtai Chen, Lizhou Dou, Yueming Zhang, Yong Liu, Shun He, Dongbing Zhao, Guiqi Wang

**Affiliations:** 1Department of Endoscopy, National Cancer Center/National Clinical Research Center for Cancer/Cancer Hospital, Chinese Academy of Medical Sciences and Peking Union Medical College, Beijing 100021, China; 2Department of Gastrointestinal Surgery, Department of General Surgery, Guangdong Provincial People’s Hospital, Guangdong Academy of Medical Sciences, Guangzhou 510080, China; 3Department of Pancreatic and Gastric Surgical Oncology, National Cancer Center/National Clinical Research Center for Cancer/Cancer Hospital, Chinese Academy of Medical Sciences and Peking Union Medical College, Beijing 100021, China; 4Department of Breast Surgical Oncology, National Cancer Center/National Clinical Research Center for Cancer/Cancer Hospital & Shenzhen Hospital, Chinese Academy of Medical Sciences and Peking Union Medical College, Shenzhen 518116, China

**Keywords:** early gastric cancer, remnant stomach, gastric tube, endoscopic submucosal dissection, radical surgery

## Abstract

(1) Background: Endoscopic submucosal dissection (ESD) for early gastric cancer (EGC) in the remnant stomach or gastric tube is not yet widespread and few studies have compared the short-term and long-term outcomes with radical surgery. (2) Methods: A total of 73 consecutive patients with EGC in the remnant stomach or gastric tube who underwent ESD or radical surgery between October 2009 and October 2020 were retrospectively analyzed in this study. Baseline characteristics, post-operative complications, quality of life (QOL), recurrence rate, overall survival (OS) and disease-free survival (DFS) were compared between the ESD and surgery groups. (3) Results: Among the 73 patients with EGC in the remnant stomach or gastric tube, 48 (65.8%) underwent ESD and 25 (34.2%) underwent surgery. The operation time (*p* = 0.000) and post-operative hospital stay (*p* = 0.002) of the ESD group were significantly shorter than those in the surgery group. The incidence of post-operative complications in the ESD group was significantly lower than that in surgery group (*p* = 0.001). The ESD group had significantly better functional scale scores and lower rates of fatigue, pain, appetite loss, financial difficulties, dysphagia, eating restrictions, hair loss, and poor body image than the surgery group. There was no significant difference in OS or DFS between the ESD and surgery groups (*p* = 0.124 and 0.344, respectively). (4) Conclusion: ESD can significantly shorten the operation time and hospital stay, reduce surgical complications, and provide better QOL for patients with EGC in the remnant stomach or gastric tube, and its long-term prognosis is no shorter than that of radical surgery.

## 1. Introduction

Gastric cancer in the remnant stomach after gastrectomy or in the gastric tube after esophagectomy is occasionally found, with an incidence of 1.0–7.0% [1,2,3,4,5,6]. Although gastric cancer in the remnant stomach or gastric tube is usually detected at an advanced stage and surgical resection of the total remnant stomach has become the standard treatment for a long time, the detection of early gastric cancer (EGC) in the remnant stomach or gastric tube is increasing because of follow-up endoscopic surveillance programs [2,7].

Endoscopic submucosal dissection (ESD) for treating EGC is less invasive than surgery and has become one of the mainstays as it can not only preserve the stomach, leading to a better quality of life (QOL), but also provide curative resection for patients without lymph node metastasis [8,9,10]. Even though ESD has already been used to treat EGC in the remnant stomach or gastric tube since 2008 [11,12], there are only a few pieces of available evidence about the long-term outcomes of ESD for EGC in the remnant stomach or gastric tube, and the results are inconsistent. Yamashina et al. [13] retrospectively analyzed the overall and cause-specific survival of patients who underwent ESD or radical surgery for EGC in the remnant stomach, and they found that the disease-specific 5-year survival rates of patients with SM2 (submucosal invasion ≥ 500 um) EGC in the ESD group was lower than in the surgery group. On the contrary, another study [14] confirmed that surgery was associated with poor survival compared with ESD for EGC in the remnant stomach. Furthermore, patients’ quality of life (QOL) can be affected by surgery. It is important that QOL be examined to see how it is influenced. However, those two studies did not compare the QOL of ESD with that of surgery for EGC in the remnant stomach, which resulted in the doubt of the reliability and benefits of ESD [13,14].

As such, in this study, we compared the short-term and long-term outcomes of ESD versus radical surgery for EGC in the remnant stomach or gastric tube, and evaluated the benefits of ESD on the improvement of QOL for those patients.

## 2. Methods

### 2.1. Patients

EGC in the remnant stomach or gastric tube is defined as a pathologically confirmed adenocarcinoma limited to the mucosal or submucosal layer of the remnant stomach or gastric tube, which is newly diagnosed after at least one year from the date of previous gastrectomy or esophagectomy [14]. Further, EGC arising in the anastomosis site (AS) is also included in this field and defined as a lesion expanding to an anastomotic site, which is classified into two types. One is defined as early gastric adenocarcinoma detected at AS more than one year after the primary operation for gastric ulcer or other non-adenocarcinoma lesions. The other one is defined as metachronous gastric cancer emerging at AS at least five years after the previous radical gastrectomy for gastric cancer, which is inconsistent with the nature and depth of invasion of the original tumor [15]. A total of 73 consecutive patients with EGC in the remnant stomach or gastric tube who underwent ESD or radical surgery at the Cancer Institute and Hospital, Chinese Academy of Medical Sciences (CICAMS), Beijing, China between October 2009 and October 2020 were identified and included in the study. The patients were divided into an ESD group and a surgery group according to the therapeutic method used. All study procedures were approved by the National Cancer Center/Cancer Hospital, Chinese Academy of Medical Science and Peking Union Medical College (Approval Number: 17-124/1380).

### 2.2. Evaluation Index

The patient baseline characteristics, and their short- and long-term outcomes were compared between the ESD and surgery groups. All gastric cancer information provided in this study was described in accordance with the Japanese Classification of Gastric Carcinoma [16]. The patient baseline characteristics included age, gender, total protein, comorbidities, information about previous surgery, and information about newly diagnosed EGC in the remnant stomach or gastric tube. We used the Charlson Comorbidity Index [17] to measure comorbidity status. Short-term outcomes encompassed the procedure time, treatment efficacy, post-operative hospital stay, and peri-operative complications. Peri-operative complications consisted of bleeding, perforation, stenosis, pleural effusion, anastomotic fistula, and wound infection, which were graded according to Clavien–Dindo’s classification [18]. Long-term outcomes contained the rates of recurrence, overall survival (OS), disease-free survival (DFS), nutritional status, and QOL after the ESD or surgery procedures. Tumor recurrence was categorized as local, regional, or distant. Local recurrence was defined as recurrences at the previous ESD site in the ESD group and at the AS in the surgery group. Regional recurrence was defined as the recurrence at the location of the peri-gastric lymph node. Distant metastasis included peritoneal carcinomatosis and metastasis to other solid organs or distant lymph nodes. Overall survival (OS) was defined as the period before death. Disease-free survival (DFS) was defined as the period before any type of recurrence. The QOL was assessed by the validated Chinese version of the European Organization for Research and Treatment of Cancer (EORTC) Quality of Life Questionnaire Core 30 (QLQ-C30) and the gastric cancer-specific module of the 22-item QOL questionnaire (QLQ-STO22) one year after the treatment or at the end of follow-up. The EORTC QLQ-C30 questionnaire consists of three scoring scales that grade function, symptoms, and global health. The EORTC-QLQ-STO22 consists of nine symptom scales. All scales were measured to a 0 to 100 linear score. A higher score represents a higher level of functioning, or a higher level of symptoms.

### 2.3. ESD Technique

All ESD procedures were performed with a standard single-channel endoscope (GIF-Q260J, Olympus) by experienced endoscopists under intravenous sedation. First, several marking dots were circumferentially made around the target lesion by using a Dual-knife (KD-650L, Olympus). Then, a saline solution mixed with epinephrine (0.002 mg/mL) and methylene blue (0.04 mg/mL) was injected into the submucosal layer to lift the lesion away from the muscle layer by entry needle (25 G, Boston Scientific, Boston, MA, USA). Finally, a circumferential mucosal incision and submucosal dissection with simultaneous hemostasis was performed by using Dual-knife and hemostatic forceps (FD-410LR, Olympus, Tokyo, Japan).

### 2.4. Surgical Technique

Radical gastrectomy with lymph node dissection was performed as standard treatment. For total residual gastrectomy, D2 was defined as a dissection of any remaining stations among 1–7, 8a, 9, 10, 11, and 12a according to the Japanese gastric cancer treatment guideline [19]. For tumors involving the gastrojejunal anastomotic site, D2 also included the dissection of the mesojejunal nodes (station J) [16].

### 2.5. Post-operative Follow-Up

Patients were evaluated every 3 months during the first 2 years after treatment, every 6 months for the following 3 years, and annually thereafter. At each visit, physical examination, blood examination (including blood routine test, blood biochemistry, and serum tumor marker assessment), endoscopic examination, and chest and abdominopelvic computed tomography imaging were performed. Information about tumor recurrence was updated every time the patient came for a follow-up visit. For those patients who did not come for a follow-up visit, data were gathered by phone calls.

### 2.6. Statistical Analysis

The independent-sample *t*-test was used to compare continuous variables that were normally distributed. Non-parametric Mann–Whitney U tests were used when the variance was not normally distributed. A Fisher’s exact test was used to compare categorical variables. Comparisons between the two groups in long-term outcomes were done by using the Kaplan–Meier method and log-rank test. A *p*-value of less than 0.05 was considered statistically significant. All tests were 2-sided and statistical analyses were performed using SPSS version 22.0 for Windows (SPSS Inc., Chicago, IL, USA).

## 3. Results

### 3.1. Baseline and Clinicopathologic Characteristics

Of the 73 patients included in the study, 48 underwent ESD (ESD group) and 25 underwent surgery (surgery group). Baseline and clinicopathological characteristics are summarized in Table 1. The patients in the ESD group were significantly older than in the surgery group (64.88 ± 8.30 vs. 57.44 ± 9.64 years, *p* = 0.001) [20]. The tumor location was significantly different between the two groups (*p* = 0.023). The differences in gender, Charlson comorbidity index, nutritional status, reasons of previous operation, type of previous operation, tumor size, macroscopic type, histologic type, lymphovascular invasion, and neural invasion between the two groups were of insignificance. Further, the clinical and pathological outcomes of the previous lesion background are presented in Appendix A.

### 3.2. Short-Term Outcomes

All the short- and long-term outcomes of ESD and surgery groups are summarized in Table 2. The operation time (39.5 ± 31.6 vs. 255.0 ± 107.8 min, respectively; *p*< 0.001) and hospital stay after treatment (6.0 ± 1.24 vs. 15.0 ± 11.07 days, respectively; *p* = 0.002) were significantly shorter in the ESD group than in the surgery group. The complication rate was significantly higher in the surgery group than in the ESD group (48.0% vs. 12.5%, *p* = 0.001). Three patients (6.25%) in the ESD group and five patients (20%) in the surgery group had a grade III–IV complication according to Clavie–Dindo classification, and were treated by emergency endoscopic management or surgery.

For 48 patients in the ESD group, the en bloc resection rate, the R0 resection rate, and the curative resection rate were 100%, 91.7%, and 73.9%, respectively. A radical surgical resection was recommended after ESD for the nine patients with noncurative resection, but it was not performed due to underlying disease or individual preference. Lymph nodes were harvested in the surgery group and only two of the twenty-five patients (8.0%) had lymph nodal metastasis. The patients with lymph nodal metastasis in the surgery group were recommended to adopt three-dimensional conformal radiotherapy combined with tegafur (S-1) administration, and no recurrence developed.

### 3.3. Long-Term Survival Outcomes

The median length of follow-up was 41.7 ± 23.29 and 73.6 ± 36.62 months in the ESD and surgery groups, respectively. The ESD group had five cases of recurrence (10.4%) and four cases of gastric cancer-related death (8.3%), while in the surgery group, there were three (12.0%) recurrences and all of them died of gastric cancer-related diseases. The clinical and pathological features of the 12 patients who died are summarized in Table 3. The long-term outcomes of ESD and surgery were analyzed by the Kaplan–Meier method and were compared by the log-rank test. There was no significant difference between the ESD and surgery groups in terms of OS (*p* = 0.124, Figure 1). The difference in DFS between the 2 groups was statistically insignificant (*p* = 0.344, Figure 2).

### 3.4. Comparison of the QOL Scores of the ESD and Surgery Groups

According to the symptoms scales of the EORTC-QLQ-C30 and the EORTC-QLQ-STO22, patients in the surgery group had significantly higher rates of fatigue (*p* < 0.001), pain (*p* = 0.048), appetite loss (*p* = 0.001), financial difficulties (*p* < 0.001), dysphagia (*p* = 0.001), eating restrictions (*p* = 0.002), hair loss (0.022), and poor body image (*p* = 0.001) than those in the ESD group (Table 4). In terms of the mean EORTC-QLQ-C30 functional scores, the ESD group had significantly better functional scales for global health status (*p* = 0.041), physical functioning (*p* = 0.010), role functioning (*p* < 0.001), emotional functioning (*p* = 0.008), cognitive functioning (*p* = 0.044), and social functioning (*p* = 0.016) compared with the surgery group.

## 4. Discussion

In the present study, we assessed the short- and long-term outcomes of patients with EGC in the remnant stomach or gastric tube who underwent ESD or radical surgery. There was no significant difference in recurrence between the two groups (10.4% vs. 12.0%, respectively). All recurrent cases were regional recurrence or distant metastasis. Several previous studies reported the long-term outcomes of ESD for EGC in the remnant stomach or gastric tube and the recurrence rate was 0–12% [13,14,21,22,23,24,25]. Fukui et al., reported the outcomes of 80 cases who underwent ESD for EGC in the remnant stomach. Twenty-five of the eighty cases underwent a noncurative resection, and tumor recurrence was observed in three patients (3/25, 12%), while no recurrence developed in patients with curative resection and radical surgery [14]. In our study, the nine patients with non-curative ESD underwent close follow-up as they refused to perform additional surgery due to underlying disease or individual preference. One of them (1/9, 11.1%) developed recurrence and died at the follow-up of 13 months, while the other eight no recurrence was observed at the end of the follow-up. The results from the previous studies on ESD for primary EGC revealed the recurrences were associated with non-curative factors such as histopathological positive resection margin, tumor size, lymphovascular invasion, or deeper tumor invasion [26,27]. However, three patients with curative resection and three patients who underwent radical surgery died of gastric cancer-related metastasis in our study. The regional recurrence and metastasis of these cases may be related to the highly malignant invasiveness of the primary tumor (one patient for stage IIIA and two patients for stage IIIC in the ESD group). Further, both the OS and DFS had no significant difference between the ESD group and radical surgery group (*p* = 0.124; *p* = 0.344, respectively), which were similar to the previous studies [5,25]. As such, the tumor recurrence and prognosis of patients who underwent ESD or radical surgery for EGC in the remnant stomach or gastric tube may be related not only to the pathological characteristics of the new early tumor, but also to the stage and invasive pathological characteristics of the primary tumor.

In our study, we used EORTC QLQ-C30 and EORTC-QLQ-STO22 to compare the effects of ESD and surgery on post-operative QOL in patients with EGC in the remnant stomach or gastric tube. We found that patients in the surgery group had significantly higher rates of fatigue, pain, appetite loss, financial difficulties, and poor body image than those in the ESD group, which may be caused by large surgical trauma and long hospital stays. Meanwhile, the rate of hair loss in patients in the surgery group was significantly higher than those in the ESD group. As we know, chemotherapy may be associated with hair loss; thus, the significant difference in hair loss in the QOL item may be due to the chemotherapy before. Further, the results of the QOL scale showed that ESD can reduce post-operation symptoms such as dysphagia and eating restriction. Just as previous studies reported, the loss of the remaining stomach strongly and broadly impaired the QOL in post-gastrectomy patients [28,29,30,31]. A possible mechanism is that the preservation of cardia or pylorus along with the residual stomach can partly prevent the pharyngeal discomfort caused by reflux, and reduce eating restriction. Thus, the presumed benefit of ESD for those patients compared to surgery can be summarized as a better post-operative QOL.

In our study, the curative resection rate was 73.9%, which was similar to the previous research reported by Satoru et al., and Byeong et al. [21,23] (78% and 71%, respectively). ESD for EGC in the remnant stomach or gastric tube is considered technically difficult due to the limited working space and tumor location involving the AS or suture line, which may result in a low curative resection rate and high complication rate such as perforation [10,12,21,27]. The complication rate was 12.5%, containing one stenosis case and five bleeding cases, and no perforation was found in the present study. However, Yohei et al., and Toshiyasu et al. [24,32] both found that the intraoperative perforation rate of ESD was high for EGC involving AS in the remnant stomach (31.4% and 50%, respectively), which is higher than that in our study. This is because we made some technical improvements: The submucosal tissue was stripped around the ANs step by step by integrating the nail and electrical knife by means of the electrical conduction effect, which can safely remove the ANs and reduce the risk of perforation [33]. In any case, the endoscopist should be more careful when performing ESD in the remnant stomach or gastric tube, to avoid perforations because of the previously mentioned anatomic features. On the other hand, the complication rate was higher in the present study than in surgery for primary EGC as reported, and the reason may be the formation of more extensive adhesions caused by the lymphadenectomy in the previous surgery [34].

As far as we know, it was the first study to compare the QOL of ESD with surgery for EGC in the remnant stomach or gastric tube. A complete follow-up was achieved in all patients in this study. In long-term cohort studies, a follow-up rate is important for reliability. Despite the small sample size, we believe the quality of the data warrants the serious consideration of our findings. However, there were still some limitations to our study. First, it was a single-center retrospective study, which may result in selection bias and the loss of some important dates. Elderly patients with serious comorbidities tended to be recommended to perform ESD, which may have influenced the overall survival rate. Second, our study involved a relatively small number of patients and a short follow-up period because of the relative rarity of EGC in the remnant stomach or gastric tube. In addition, this study was the first to consider the risk factors of the previous lesions, such as the type of cancer or tumor TNM stage, on the long-term prognosis of EGC in the remnant stomach or gastric tube. However, the lack of clinical and pathological information on the previous lesions in some cases also makes it impossible for us to conduct multivariate analysis. Finally, the QOL assessment scale may have a subjective element and the scores may be influenced by the obtaining time and approach. Thus, confirmation studies with a larger multi-institutional population and adequate follow-up duration are required to confirm the safety, efficacy, and suitable criteria of ESD for EGC in the remnant stomach or gastric tube.

In conclusion, for patients with EGC in the remnant stomach or gastric tube, receiving ESD had better peri-operative outcomes in terms of operative time, hospital stay, nutritional status, and QOL compared with radical surgery. Meanwhile, from the results of our study, there was no significant difference in recurrence rate and mortality between radical surgery and ESD, but the long-term prognosis was strongly influenced by the initial treatment. ESD could be used as an additional treatment for patients with EGC confined to the superficial submucosa in the residual stomach and elderly patients with poor basic physical conditions who cannot tolerate surgical treatment. Such patients could benefit more from ESD than surgery. The present results might provide endoscopists with useful information for preprocedural decision-making for EGC in the remnant stomach or gastric tube.

## Figures and Tables

**Figure 1 jcm-11-05403-f001:**
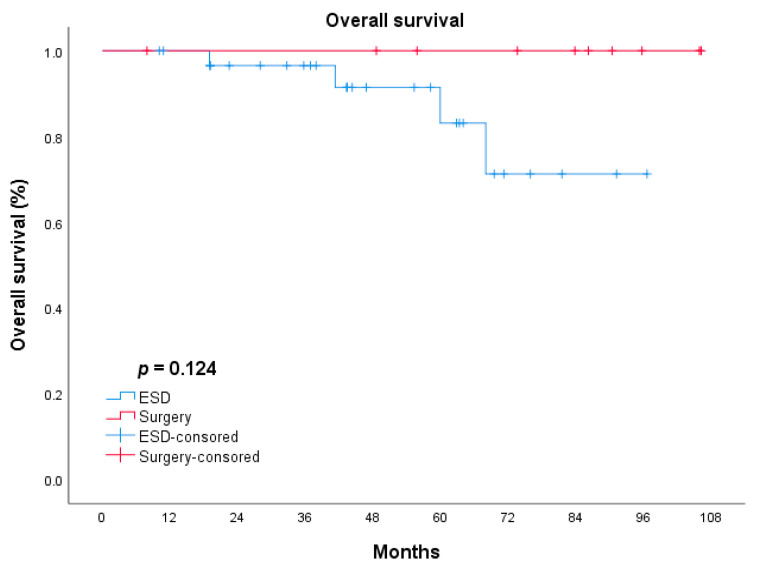
OS of patients in ESD and surgery groups. Log-rank test: *p* = 0.496.

**Figure 2 jcm-11-05403-f002:**
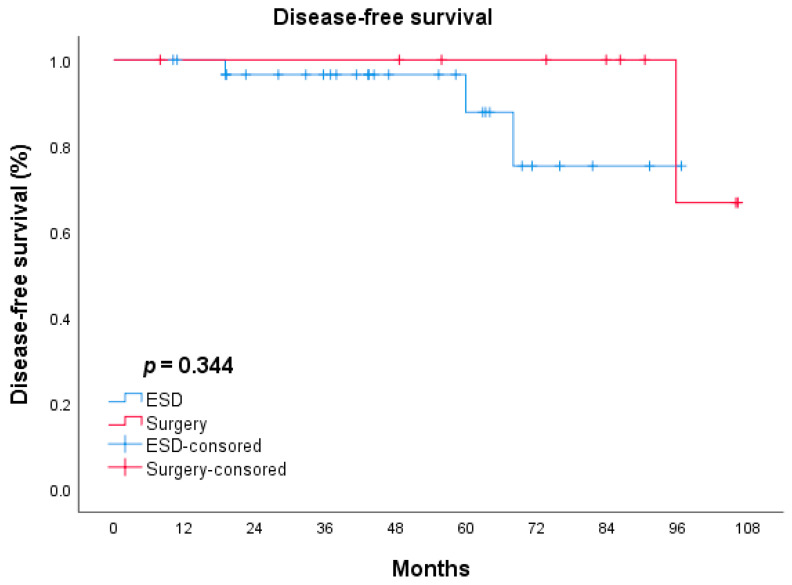
DFS of patients in ESD and surgery groups. Log-rank test: *p* = 0.554.

**Table 1 jcm-11-05403-t001:** The characteristics and clinical data of the patients.

	ESD (n = 48)	Surgery (n = 25)	*p* Value
**Gender, number (%)**			0.295
Male	43 (89.6%)	20(80.0%)	
Female	5 (10.4%)	5(20.0%)	
**Age, mean (SD), y**	64.88 (8.30)	57.44(9.64)	0.001
**Reasons of previous operation, number (%)**			0.786
Gastric cancer	36 (75.0%)	17(68.0%)	
Benign gastric ulcer	1 (2.1%)	6(24.0%)	
Esophageal cancer	10 (20.8%)	0(0%)	
Other (mesenchymoma/leiomyoma)	1 (2.1%)	2 (8.0%)	
**Type of previous operation, n (%)**			0.037
Distal gastrectomy			
Billroth-I	14 (29.2%)	14(56.0%)	
Billroth-II	9 (18.8%)	5(20.0%)	
Proximal gastrectomy	15 (31.3%)	6(24.0%)	
Esophagectomy	10 (20.8%)	0(0%)	
**Tumor location, n (%)**			0.023
Fundus	1 (2.1%)	1(4.0%)	
Body	14 (29.2%)	8(32.0%)	
Antrum	15 (31.3%)	2(8.0%)	
Cardia	17 (35.4%)	10(40.0%)	
Anastomosis site	1 (2.1%)	4(16.0%)	
**Tumor size, mm**			
Median (P25,P75)	20.00 (10.25, 30.00)	25.00(18.50,35.00)	0.077
**Macroscopic type, n (%)**			0.097
0-I	4 (8.4%)	10(40.0%)	
0-IIa	15 (31.3%)	1(4.0%)	
0-IIb	3 (6.3%)	5(20.0%)	
0-IIc	7 (14.6%)	3(12.0%)	
0-IIc + IIa	19 (39.6%)	3(24.0%)	
**Histologic type, n (%)**			0.016
Differentiated	34 (70.8%)	12(48.0%)	
Undifferentiated	14 (29.2%)	13(52.0%)	
**Depth of tumor invasion, n (%)**			0.008
M	40 (83.4%)	17 (68%)	
SM1	4 (8.3%)	2 (8.0%)	
SM2	4 (8.3%)	6 (24.0%)	
**Lymphovascular invasion, n (%)**			0.331
Present	2 (4.2%)	3 (12.0%)	
Absent	46 (95.8%)	22 (88.0%)	
**Neural invasion, n (%)**			0.547
Present	1 (2.1%)	2 (8.0%)	
Absent	47 (97.9%)	23 (92.0%)	

BMI, body mass index; SD, standard deviation; M, mucosa; SM, submucosa.

**Table 2 jcm-11-05403-t002:** Short- and long-term outcomes of ESD and surgery groups.

	ESD (n = 48)	Surgery (n = 25)	*p* Value	
**Short-term outcomes**				
**Operation time, median (SD), minute**	39.5 (31.6)	255.0 (107.8)	0.000	
**Hospital stays after surgery, median (SD), day**	6.0 (1.24)	15.0 (11.07)	0.002	
**Peri-operative complication, number (%)**	6 (12.5%)	12 (48%)	0.001	
bleeding	5	1		
perforation	0	0		
stenosis	1	-		
pleural effusion	-	1		
anastomotic fistula	-	4		
wound infection	-	3		
Other	-	3		
**Clavien** **–** **Dindo, number (%)**				
I–II	3 (6.25%)	7 (28%)		
III–IV	3 (6.25%)	5 (20%)		
V	0	0		
**Treatment efficacy, number (%)**				
R0 resection	44 (91.7%)	-	-	
En bloc resection	48 (100%)	-		
Curative resection	39 (73.9%)	-		
Lymph node metastasis	-	2 (8.0%)	-	
**Long-term outcomes**				
**Length of follow-up, median (SD), month**	41.7 (23.29)	73.6 (36.62)	0.684	
**Recurrence, number (%)**	5 (10.4%)	3 (12.0%)		
**Gastric cancer-related deaths, number (%)**	4 (8.3%)	3 (12.0%)		
**Time after treatment to recurrence, median (SD), month**	36.0 (78.56)	69.0 (142.48)		

ESD, Endoscopic submucosal dissection; SD, standard deviation.

**Table 3 jcm-11-05403-t003:** Clinical and pathological features and outcomes of the dead during follow-up.

Case	Age	Reason of Previous Operation	TNM Staging of Previous Lesions	Type of Secondary Operation	Tumor Characteristics						Location of Recurrence	Additional Treatment for Recurrence	Cause of Death
					Macroscopic Type	Histological Type	Depth of Tumor Invasion	Lymphovascular Invasion	Neural Invasion	Curative Resection			
1	67	Esophageal cancer	IB	ESD	II-a + II-c	Differentiated	Mucosa	Negative	Negative	Yes	Bone	Symptomatic treatment	Esophageal cancer-related
2	58	Gastric cancer	IIIA	ESD	II-b	Differentiated	Mucosa	Negative	Negative	Yes	Liver	Chemoradiotherapy	Gastric cancer-related
3	56	Gastric cancer	IIIC	ESD	II-a + II-c	Differentiated	Mucosa	Negative	Negative	Yes	Colon	Symptomatic treatment	Gastric cancer-related
4	74	Gastric cancer	NA	ESD	I-s	Differentiated	Mucosa	Negative	Negative	Yes	-	-	Colon cancer-related
5	62	Gastric cancer	IIIC	ESD	II-a + II-c	Undifferentiated	Mucosa	Negative	Negative	Yes	Bone	Radiotherapy	Gastric cancer-related
6	77	Gastric cancer	NA	ESD	II-a	Differentiated	Mucosa	Negative	Negative	Yes	-	-	Apastia
7	58	Esophageal stromal tumor	NA	ESD	II-c	Undifferentiated	Mucosa	Negative	Negative	No	Liver and retroperitoneal lymph node	Symptomatic treatment	Gastric cancer-related
8	57	Gastric ulcer	-	TG	I-s	Undifferentiated	Submucosa	Negative	Positive	-	Liver and retroperitoneal lymph node	Chemoradiotherapy	Gastric cancer-related
9	71	Gastric ulcer	-	TG	II-b	Differentiated	Mucosa	Negative	Negative	-	-	-	Myocardial infarction
10	73	Gastric cancer	NA	TG	I-s	Differentiated	Mucosa	Negative	Negative	-	Liver and bone	Symptomatic treatment	Gastric cancer-related
11	57	Gastric cancer	NA	TG	I-s	Differentiated	Mucosa	Negative	Negative	-	-	-	Esophageal cancer-related
12	45	Gastric cancer	NA	TG	II-c	Undifferentiated	Mucosa	Negative	Negative	-	Liver and retroperitoneal lymph node	Surgery and Chemoradiotherapy	Gastric cancer-related

NA, Not available; TG, Total gastrectomy; ESD, Endoscopic submucosal dissection.

**Table 4 jcm-11-05403-t004:** Functional and symptom scales of the EORTC-QLQ-C30 and the EORTC-QLQ-STO22.

	ESD (n = 32)	Surgery (n = 13)	*p*-Value
**EORTC-QLQ-C30** **functional scales, mean (SD)**			
Global health status	71.58 (15.06)	61.90 (12.54)	0.041
Physical functioning	97.71 (5.25)	84.29 (16.46)	0.010
Role functioning	95.83 (11.20)	69.05 (14.41)	<0.001
Emotional functioning	91.41 (16.87)	74.40 (23.68)	0.008
Cognitive functioning	97.92 (7.02)	89.29 (14.03)	0.044
Social functioning	94.79 (14.93)	76.19 (24.21)	0.016
**EORTC-QLQ-C30** **symptom scales, mean (SD)**			
Fatigue	6.25 (13.22)	25.40 (17.66)	<0.001
Nausea and vomiting	7.81 (15.25)	14.29 (11.05)	0.160
Pain	3.65 (9.21)	10.71 (14.03)	0.048
Dyspnea	11.46 (42.00)	11.90 (16.57)	0.970
Insomnia	13.54 (25.20)	23.81 (30.46)	0.239
Appetite loss	2.08 (8.20)	45.24 (38.36)	0.001
Constipation	14.58 (26.69)	28.57 (25.68)	0.105
Diarrhea	13.54 (22.17)	23.81 (24.21)	0.167
Financial difficulties	5.21 (12.30)	28.57 (22.10)	<0.001
**EORTC-QLQ-STO22** **symptom scales, mean (SD)**			
Dysphagia	4.86 (7.95)	20.63 (12.97)	0.001
Pain	7.55 (11.07)	27.98 (22.55)	<0.001
Reflux symptoms	15.63 (18.69)	19.84 (15.21)	0.462
Eating restrictions	3.91 (7.92)	23.81 (19.30)	0.002
Anxiety	5.90 (15.71)	22.22 (27.91)	0.056
Dry mouth	3.13 (9.87)	11.90 (16.57)	0.082
Taste	3.13 (13.01)	9.52 (15.63)	0.156
Hair loss	2.08 (8.20)	14.29 (17.12)	0.022
Body image	2.08 (8.20)	28.57 (22.01)	0.001

EORTC, European Organization for Research and Treatment of Cancer; QLQ-C30, Quality of Life Questionnaire Core 30; QLQ-STO22, 22-item Quality of Life Questionnaire; SD, standard deviation.

## Data Availability

The data that support the findings of this study are available from the corresponding author, upon reasonable request.

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
