# Peer review of "Comparison of Endoscopic Submucosal Dissection and Radical Surgery for Early Gastric Cancer in Remnant Stomach"

_jcm, 2022, doi:10.3390/jcm11185403_

Round 1
Reviewer 1 Report
This article comparing ESD to radical surgery as treatment methods for gastric cancer arising in the residual stomach. As a novelty, the paper emphasizes that the quality of life of both groups was examined.
Short-term outcomes showed significantly shorter treatment time for ESD, but the treatment time was too short, so there is likely to be a significant bias in the choice between ESD and radical surgery.
1. Regarding the patient's choice, why did you choose radical surgery instead of ESD for intramucosal carcinoma?
2. The size of the lesion is not listed. Please describe the size. Is it possible that the choice of treatment was based on size?
3. Long-term survival outcomes are likely to be determined by the results of initial treatment, but in one-third of cases the TMN stage is unknown, making it difficult to compare the two groups in terms of long-term prognosis. Please consider whether to keep these results (OS and DFS).
4. The significant difference in hair loss in the QOL item may be due to the fact that 12 of the 14 patients in the radical surgery group who required additional treatment received chemotherapy, which is not an item that should be compared to the ESD group.
5. There is an error in the number of patients on line 220.
Reviewer 2 Report
Important research area, with some comments:
Line 186-187: On what basis the patients allocated to each group of
therapy??
Is there is early at that time 2010, expeets in ESD? or it is a
training time??
Line 245-246: Twenty-five of the 80 cases underwent a noncurative resection 246 and tumor recurrence was observed in 3 patients (3/25, 12%)!!!! what about the other 22 cases???
Line 250-252: One of them (1/9, 250 11.1%) developed recurrence and died at the follow-up of 13 months, while the other 251 eight were not observed recurrence at the end of the follow-up What is your explanation of this phenomenon???
Statistical analysis: you have to add sample size calculation.
Table 1: Reasons of previous operation: Gastric ulcer. What type of ulcer!!??
Other. What you mean!!??
Better to add abbreviations as a separate title.
Round 2
Reviewer 1 Report
Dear author's,
You responded appropriately to my requested comments. The size of the lesions reveals the background of the patients.
After reading this article, I understand that ESD as an additional treatment for the residual stomach has a shorter treatment time and better quality of life than radical surgery, but the long-term prognosis is strongly influenced by the initial treatment. At the end of the paper (Discussion paragraph), please state the author's opinion on which patients you would recommend ESD.
